# Mapping of promoter usage QTL using RNA-seq data reveals their contributions to complex traits

**Naoto Kubota**, **Mikita Suyama** *

Division of Bioinformatics, Medical Institute of Bioregulation, Kyushu University, Fukuoka, Japan

* mikita@bioreg.kyushu-u.ac.jp

**Data Availability Statement:** All codes used in this study is available at https://github.com/NaotoKubota/Kubota_puQTL.

**Funding:** The authors received no specific funding for this work.

## Abstract

Genomic variations are associated with gene expression levels, which are called expression quantitative trait loci (eQTL). Most eQTL may affect the total gene expression levels by regulating transcriptional activities of a specific promoter. However, the direct exploration of genomic loci associated with promoter activities using RNA-seq data has been challenging because eQTL analyses treat the total expression levels estimated by summing those of all isoforms transcribed from distinct promoters. Here we propose a new method for identifying genomic loci associated with promoter activities, called promoter usage quantitative trait loci (puQTL), using conventional RNA-seq data. By leveraging public RNA-seq datasets from the lymphoblastoid cell lines of 438 individuals from the GEUVADIS project, we obtained promoter activity estimates and mapped 2,592 puQTL at the 10% FDR level. The results of puQTL mapping enabled us to interpret the manner in which genomic variations regulate gene expression. We found that 310 puQTL genes (16.1%) were not detected by eQTL analysis, suggesting that our pipeline can identify novel variant–gene associations. Furthermore, we identified genomic loci associated with the activity of "hidden" promoters, which the standard eQTL studies have ignored. We found that most puQTL signals were concordant with at least one genome-wide association study (GWAS) signal, enabling novel interpretations of the molecular mechanisms of complex traits. Our results emphasize the importance of the re-analysis of public RNA-seq datasets to obtain novel insights into gene regulation by genomic variations and their contributions to complex traits.

## Author summary

Many variations exist in the human genome, creating phenotypic diversity among individuals. It is well known that they are associated with the risk of disease development by affecting the expression levels of genes. Genes are transcribed from regulatory elements called promoters. Although some genes are transcribed from multiple promoters and translated into proteins with different functions, the relationship between genomic variations and promoter activities has not been investigated in depth compared to the relationship between genomic variations and gene expression levels. In this study, we proposed a

**Competing interests:** The authors have declared that no competing interests exist.

new method to detect the association between genomic variations and promoter activities. Our method identified the associations between many variations and promoters using genomic and promoter activity data from blood cells of 438 individuals. This study allowed us to identify new functional associations between genomic variations and genes. Furthermore, we identified previously undiscovered variation-gene-disease associations. Our results will help to elucidate the molecular mechanisms of diseases in which genetic factors are involved.

## Introduction

Variations in the human genome generate a variety of transcriptomes across individuals. Many previous studies have identified single-nucleotide polymorphisms (SNPs) and short insertions and deletions (indels) associated with gene expression levels, which are called expression quantitative trait loci (eQTL) [1]. eQTL are significantly enriched in genomic loci associated with complex traits identified by genome-wide association studies (GWASs), suggesting the importance of the differences in gene regulation caused by genomic variations in the context of various traits and disease risk [2]. In conventional eQTL studies, the expression levels of all isoforms of each gene are quantified, and they are then merged into the total gene expression levels and used as dependent variables in linear regression analyses [1]. If a gene has multiple promoters, it is impossible to directly assess the effects of genomic variations on the activity of each promoter using eQTL analyses. Isoforms transcribed from distinct promoters can have distinct functions even if they have the same gene name; for example *TP53* (which encodes p53) [3], *TP73* (p73) [4,5], *SHC1* (p52/p46/p66) [6,7], *INK4a/ARF* [8,9], and *CDKN1A* (p21) [10]. Therefore, it is essential to map the genomic variations associated with promoter activity to better understand how they affect the expression levels of specific transcripts and confer risk for complex traits.

Recently, several studies mapped the genomic variations associated with promoter activity based on cap analysis of gene expression (CAGE) technology, which captures the 5′ end cap structure of transcripts and is termed promoter usage QTL (puQTL) [11,12] or transcription start site QTL (tssQTL) [13]. Those studies have shown that puQTL or tssQTL analysis enables the detection of SNPs associated with promoter usage, which conventional eQTL analysis could not discover. However, this strategy has a high cost because it requires the generation of promoter usage data for hundreds of individuals. Thus, for mapping additional puQTL at a lower cost, it is necessary to develop a new method that can leverage extensive RNA-seq data from public databases.

Several RNA-seq-based studies have been performed to map transcript usage QTL (tuQTL) [14–18]. Most of those studies were based on transcript annotations; thus, promoters not described in standard annotation files could not be analyzed. Notably, a recent study [14] reported a computational method to map tuQTL by splitting data into independent events (promoter usage, splicing, and 3′ end usage), and employed Salmon [19] for quantifying full-length transcripts. Alignment-free methods, such as Salmon and kallisto [20], have been reported to have a low performance in transcript-level quantification [21] and to overestimate lowly expressed transcripts [22], which might lead to the mapping of false-positive puQTL. Therefore, a method that focuses on promoter activity estimates and overcomes the problems caused by the alignment-free methods is essential for accurately mapping puQTL using RNA-seq data.

This study aimed to develop a new method to map puQTL by quantifying genome-wide promoter activities using publicly available RNA-seq data. In our pipeline, we employed an alignment-based method, proActiv, which has shown high performance in promoter activity estimates, as well as higher levels of agreement with H3K4me3 ChIP-seq signals compared with other methods, such as Salmon and kallisto [23]. By newly constructing transcript annotations based on mapped reads, we tried to comprehensively capture the activities of promoters, including those not described in standard annotation files. Moreover, in addition to short variants (SNPs and indels), we explored associations between long variants (structural variants (SVs)) and promoter usage. We successfully mapped puQTL and found that most of them had genomic and epigenetic features that were similar to those of eQTL, in agreement with a previous report [12]. We also found that our methodology was able to detect hidden promoters associated with genomic variations, which may have been overlooked by annotation-based puQTL analyses. Some puQTL were colocalized with GWAS signals, which enabled new interpretations of complex trait associations. Overall, this study provided a successful way to identify genetic factors that regulate promoter activity by leveraging publicly available RNA-seq data. We can expand this strategy to extensive RNA-seq datasets from various tissues and cells, thus advancing our understanding of the context-specific perturbation of transcription by genomic variations.

## Methods

### Preparation of genotype and RNA-seq data

We obtained metadata and genotype data of SNPs and indels for 438 individuals from the 1000 Genomes Project [24] in variant call format (vcf) (hg19). Subsequently, we transformed these coordinates into hg38 coordinates using UCSC liftOver. As for SVs, we downloaded the genotype data reported by Yan et al. in vcf (hg38) [25]. Then, we combined them and used the genotype data of biallelic variants with a minor allele frequency >0.01 in the samples, for downstream analysis. We obtained corresponding RNA-seq data of Epstein–Barr virus-transformed lymphoblastoid cell lines (LCLs) from the GEUVADIS project [17].

### RNA-seq data processing

We performed the quality control of RNA-seq data using fastp (version 0.20.1) [26] with "-3 -q 30" options, to discard reads with low quality. We aligned the remaining reads to the human reference genome (GRCh38) and the Ensembl 104 transcript annotations using STAR (version 2.7.9a) [27] in the two-pass mode. In the first mapping step, we used the "--outFilterMultimapNmax 1" option to allow uniquely mapped reads exclusively. In the second mapping step, we used the "--outFilterMultimapNmax 1--sjdbFileChrStartEnd /path/to/1_SJ.out.tab /path/to/2_SJ.out.tab . . .. /path/to/438_SJ.out.tab--limitSjdbInsertNsj 10000000" option to use the junctions files of the first mapping as "annotated" junctions for the second mapping step. These steps allowed us to detect more spliced reads mapped to novel junctions. Using the mapped reads, we performed reference-guided transcript assembly using StringTie2 (version 2.1.7) [28] with the "-G /path/to/Ensembl_104_annotations.gtf--conservative" option for each sample, and merged them with the "-G /path/to/Ensembl_104_annotations.gtf--merge" option. Using the transcript annotations file and junction counts files produced by STAR, we quantified and normalized the promoter activities for each sample using proActiv (version 1.1.18) [23]. The proActiv software is based on the concept of weighted splicing graphs to define unique identifiable promoters. It performs normalization of total junction reads using DESeq2, which is well used software for normalization of gene expression, and log2 transformation to obtain absolute promoter activity. We executed functions in the proActiv software

in a docker container (naotokubota/proactiv:1.1.18 in Docker Hub). The promoters of single-exon genes and those that overlapped with internal exons were excluded. All promoters with promoter activity greater than zero in at least 25% of the samples were used for puQTL analysis. For the quantification of gene-level expressions, we counted reads for each transcript using featureCounts (version 1.6.4) [29] with the "-p -B -t exon -g gene_id -a /path/to/transcript_annotations_assembled_by_StringTie2.gtf" option. In accordance with the methods employed in the GTEx project [2], all genes with a transcripts per million (TPM) value >0.1 and raw read counts greater than six in at least 20% of the samples were used for eQTL analysis. The read counts of the remaining genes were normalized across samples via the trimmed mean of M values method implemented in edgeR (version 3.12.1) [30] in a docker container (broadinstitute/gtex_eqtl:V8 in Docker Hub).

We downloaded the gene quantification data of the GM12878 cell line generated by CAGE data processing from the ENCODE project (Accession ID: ENCFF006DIB) to compare the promoter activities estimated by proActiv [23] and kallisto [20]. We downloaded GM12878 RNA-seq data from the ENCODE project (Accession ID: ENCFF000EWJ, ENCFF000EWX). After quality control using fastp with the "-3 -q 30" option, we performed a random sampling of 50 million reads using SeqKit (version 0.13.2) [31] with the "sample -n 50000000 -s 20" option. We obtained the TPM value for each transcript based on the remaining reads using kallisto (version 0.44.0) and summed those of transcripts sharing the same promoter. We also obtained promoter activity scores using proActiv, as well as GEUVADIS RNA-seq data processing. We excluded promoters with zero activity in all methods (CAGE, kallisto, and proActiv) and calculated Spearman's $\rho$ for each pair of promoter activity sets.

## QTL mapping

We calculated probabilistic estimation of expression residuals (PEER) factors [32] based on the promoter activity and gene expression matrix for the puQTL and eQTL analysis using the PEER package (version 1.0) in a docker container (broadinstitute/gtex_eqtl:V8 in Docker Hub). For QTL discovery, we used QTLtools [33] (version 1.3.1) in a docker container (naotokubota/qtltools:1.3.1 in Docker Hub). We calculated genetic principal components (PCs) from the genotype matrix using QTLtools pca with the "--scale--center--maf 0.01--distance 50000" option. First, we used the nominal pass using QTLtools cis with the "--nominal 0.00001--normal" option, to check the suitable number of PEER factors (as covariates) that were necessary to improve sensitivity. We checked the number of QTL with nominal $P$-values $<1.0 \times 10^{-5}$ using varying numbers of PEER factors (0, 1, 2, 3, 4, 5, 6, 7, 8, 9, 10, 15, 20, 25, 30, 35, 40, 45, and 50). Next, we performed a permutation pass using QTLtools cis with the "--permute 1000--normal--std-err" option. We included the top five genetic PCs, sex, and the suitable number of PEER factors for each category (25 for puQTL and 50 for eQTL) as covariates in the QTL analysis. Subsequently, using the output of the permutation pass, we performed a false discovery rate (FDR) correction on the permutation $P$-values at the 10% FDR level using "Rscript /qtltools/scripts/qtltools_runFDR_cis.R." Finally, we performed a conditional pass to obtain the significant QTL at the 10% FDR level using QTLtools cis with the "--mapping /path/to/thresholds.txt--normal--std-err" option. We also performed a nominal pass using QTLtools cis with the "--nominal 1.0--normal--std-err" option, to obtain the statistics of all associations in a ±1 Mb *cis* window. To estimate causal variants for puQTL, we performed fine-mapping using SuSiE (version 0.11.42) [34,35] in a docker container (naotokubota/coloclocuscomparer:1.0 in Docker Hub). We obtained the posterior inclusion probability (PIP) for each variant–phenotype pair using the "susie_rss()" function. We used the UCSC Genome Browser [36] to visualize the genomic positions around the QTL.

## HiChIP data processing

We downloaded the published data of H3K27ac HiChIP of the GM12878 cell line (GEO accession number: GSM2705041) [37] and performed quality control using fastp (version 0.20.1) [26] with the "-3 -q 30" option, to discard reads with low quality, as well as RNA-seq data processing. We aligned the remaining reads to the human reference genome (GRCh38) using the HiC-Pro pipeline (version 3.0.0) [38] in a docker container (nservant/hicpro:3.0.0 in Docker Hub). We used the default settings to remove duplicate reads, assign reads to MboI restriction fragments, and filter valid interactions. We performed peak calling using MACS2 [39] with the "callpeak -f BAM -g hs -q 0.01" option and obtained sets of high-confidence loops (5 kb bin, 1% FDR level) using FitHiChIP (version 7.0) [40] with the configfile_BiasCorrection_CoverageBias file and default settings in a docker container (aylab/fithichip:latest in Docker Hub).

## Genomic and epigenetic enrichment analysis

We downloaded the peak and signal files of ChIP-seq of histone marks and ATAC-seq of the GM12878 cell line from the ENCODE project [41] (Accession ID: H3K4me3, ENCFF587DVA; H3K4me1, ENCFF321BVG; H3K27ac, ENCFF023LTU; H3K9ac, ENCFF069KAG; H3K27me3, ENCFF291DHI; ATAC-seq, ENCFF748UZH), transcription factor (TF) footprinting data based on DNase-seq of the GM12865 and GM06990 cell lines [42], and the chromatin states data of the GM12878 cell line defined by ChromHMM [43] from the Roadmap Epigenomics Project [44]. The chromatin states fell into 25 categories: 1_TssA (Active TSS), 2_PromU (Promoter Upstream TSS), 3_PromD1 (Promoter Downstream TSS 1), 4_PromD2 (Promoter Downstream TSS 2), 5_Tx5'' (Transcribed--5' preferential) 6_Tx (Strong transcription), 7_Tx3'' (Transcribed--3' preferential), 8_TxWk (Weak transcription), 9_TxReg (Transcribed & regulatory (Prom/Enh)), 10_TxEnh5'' (Transcribed 5' preferential and Enh), 11_TxEnh3'' (Transcribed 3' preferential and Enh), 12_TxEnhW (Transcribed and Weak Enhancer), 13_EnhA1 (Active Enhancer 1), 14_EnhA2 (Active Enhancer 2), 15_EnhAF (Active Enhancer Flank), 16_EnhW1 (Weak Enhancer 1), 17_EnhW2 (Weak Enhancer 2), 18_EnhAc (Primary H3K27ac possible Enhancer), 19_DNase (Primary DNase), 20_ZNF/Rpts (ZNF genes & repeats), 21_Het (Heterochromatin), 22_PromP (Poised Promoter), 23_PromBiv (Bivalent Promoter), 24_ReprPC (Repressed Polycomb), and 25_Quies (Quiescent/Low). Moreover, we tested whether a set of QTL is significantly enriched in the functional annotations using QTLtools fenrich with the "--permute 1000" option. We used the positions of best hit variants for each QTL and those of all promoters ($n$ = 12,957) and all genes ($n$ = 12,275) for the QTLtools fenrich analysis. We generated aggregation plots of the ChIP-seq and ATAC-seq profiles of the GM12878 cell line (Accession ID: H3K4me3, ENCFF927KAJ; H3K4me1, ENCFF564KBE; H3K27ac, ENCFF469WVA; H3K9ac, ENCFF028KBY; H3K27me3, ENCFF919DOR; ATAC-seq, ENCFF603BJO) using deepTools (version 3.5.1) [45] in a docker container (quay.io/biocontainers/deeptools:3.5.1--py_0 in Quay.io) with the positional information of the significant QTL in the bed format.

## GWAS hit enrichment and colocalization analysis

We tested whether the QTL variants are concordant with GWAS lead variants using QTLtools rtc with the "--hotspots /path/to/hotspots.bed --normal --gwas-cis" option. We downloaded the recombination hotspot data from the 1000 Genomes Project and GWAS lead variant data from the GWAS Catalog [46] for the QTLtools rtc analysis. We also performed a colocalization analysis using coloc (version 5.1.0) [47] in a docker container (naotokubota/coloc-locuscomparer:1.0 in Docker Hub) with the GWAS summary statistics obtained from the GWAS Catalog. We calculated $r$-squared values ($r^2$) between variants of interest and variants around them

using PLINK (version 1.90b6.21) [48] with the "--r2--ld-window-kb 1000--ld-window 99999--ld-window-r2 0" option in a docker container (quay.io/biocontainers/ plink:1.90b6.21--h779adbc_1 in a Quay.io).

## Comparison of multiple protein sequences

We downloaded the protein sequences of interest from UniProt (https://www.uniprot.org) and performed multiple alignments of protein sequences using MUSCLE (https://www.ebi.ac. uk/Tools/msa/muscle/) [49]. We searched for protein domains in the sequences using InterPro (https://www.ebi.ac.uk/interpro/) [50]. To predict protein structures from the sequences, we used ColabFold [51], which is a free platform of AlphaFold2 coupled with Google Colaboratory.

## Computational environments

The computational pipeline was developed in the GNU Bash 3.2 environment. We generated all graphs using the pandas (version 1.1.3), matplotlib (version 3.3.1), and seaborn (version 0.11.0) packages in the Python environment (version 3.8.5). Furthermore, we used the scipy (version 1.6.2) package for all statistical tests, and docker (version 20.10.7) for the stable execution of various software.

## Results

### Systematic identification of puQTL using RNA-seq data

We aimed to identify genomic variations associated with promoter activities (puQTL) and total gene expression levels (eQTL) using RNA-seq data (Fig 1A). First, we checked whether it is possible to estimate promoter activity accurately from RNA-seq data. We compared the promoter activity of the GM12878 cell line measured by CAGE with those estimated by proActiv [23] and kallisto [20], which can accept RNA-seq data as input. Our results also showed that the promoter activity estimated by proActiv was correlated more strongly with that measured by CAGE vs. kallisto (Spearman's $\rho$ = 0.647 and 0.432 for proActiv and kallisto, respectively) (S1 Fig). The proActiv software has been reported to exhibit high performance in promoter usage inference compared with other methods, and to capture changes in alternative promoter usage in cancer transcriptomes [23]. Although proActiv is limited in that it cannot estimate the activities of promoters that overlap with internal exons, we obtained a considerable advantage in accuracy and confirmed that it is possible to utilize publicly available RNA-seq data from various resources; thus, we employed proActiv for the estimation of promoter activity in this study. Next, we developed computational pipelines for puQTL and eQTL identification using the RNA-seq dataset of LCLs from 438 individuals provided by the GEUVADIS project [17] (Figs 1B and S2 and S1 Table). After quality control, we aligned the RNA-seq reads to the human reference genome in the two-pass mode implemented in STAR to obtain as many junction reads as possible, because the estimation of promoter activity by proActiv depends on the number of junction reads mapped on the first and second exon. Using the mapped reads of 438 individuals and the Ensembl gene annotation file, which describes the structure of all genes including coding and non-coding, we re-constructed gene annotations using StringTie2 as an input file of proActiv. These steps allowed us to gain additional promoters to be analyzed, because proActiv can only detect the promoters described in the input GTF file. Using datasets of normalized promoter activities, gene expression levels, and genotypes, we mapped puQTL and eQTL using QTLtools. The numbers of input phenotypes for puQTL (promoters) and eQTL (genes) mapping were 12,957 and 12,275, respectively. To maximize the number of

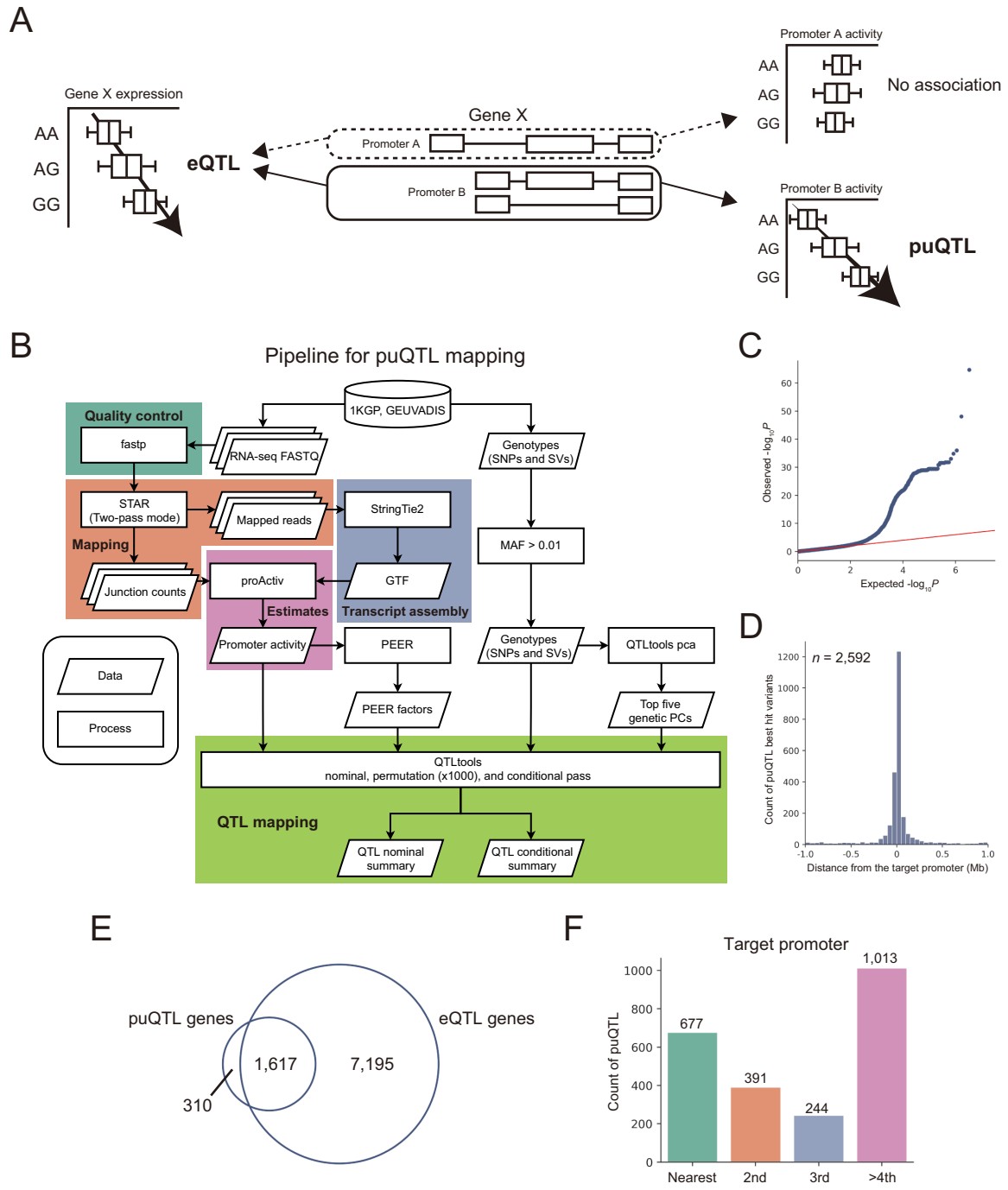

**Fig 1. puQTL mapping using RNA-seq data.** (A) Schematic view of eQTL and puQTL mapping. (B) Pipeline for puQTL mapping. 1KGP, the 1000 Genomes Project; MAF, minor allele frequency; GTF, gene transfer format. (C) Quantile–quantile plot of P-values. The nominal pass results of chromosome 22 are plotted, and the red line indicates the expected P-values under the null hypothesis. (D) Distribution of the distance of puQTL best hit variants from the target promoters. (E) Overlap of puQTL genes and eQTL genes. (F) Counts of puQTL associated with multi-promoter genes based on categories of proximity to their target promoters. "Nearest" means that the puQTL targets the nearest promoters of the associated genes, "2nd" means that they target the second nearest promoters, and so forth.

significant puQTL signals, we used the top five genetic PCs, sex, and PEER factors [32] calculated based on phenotype tables as covariates in the regression analysis (S3 Fig). To conduct a comprehensive survey of the genetic factors associated with gene regulation, here we used genotypes of not only short variants, such as SNPs and indels, but also SVs recently discovered by combined analysis of short-read and long-read sequencing data [25]. As a result, we successfully mapped 2,592 puQTL and 18,205 eQTL at the 10% FDR level. One reason for the difference in total number of signals between puQTL and eQTL is probably the limited power to estimate promoter activity. While all reads mapped to exons are used to quantify gene expression levels (eQTL), only junction reads spanning the first and second exons are used to estimate promoter activity (puQTL). This reduces the number of reads that can be used to quantify, reducing the accuracy of the promoter activity estimation and the power of mapping QTL. We observed substantial deviations of the resulting *P*-values from the null expectation for both puQTL and eQTL, indicating that our analysis was well calibrated (Figs 1C and S4A). We found that puQTL and eQTL were likely to be located near target promoters (Figs 1D and S4B), in concordance with previous reports [12]. Moreover, although we encountered a limitation in that internal promoters were excluded from our analysis, we confirmed that our results were consistent with those of the CAGE-based puQTL analysis [12]; i.e., an intergenic variant, rs8028374 (A/G), was mapped as puQTL associated with the most external promoters of the *TTC23* gene, prmtr.35339 ($P = 6.99 \times 10^{-18}$, β = −0.56) (S5A and S5B Fig), and an exonic variant, rs35430374 (C/A), was not mapped as puQTL associated with external promoters of the *DENND2D* gene ($P = 7.78 \times 10^{-5}$, β = −0.41 for prmtr.53803; $P = 0.22$, β = −0.13 for prmtr.53804) (S5C and S5D Fig). Although the number of puQTL was smaller than those of the previous CAGE-based puQTL analysis because of the limited power of RNA-seq-based estimates of promoter activities, these results demonstrated that our pipeline provides promising results. We found that 16.1% of the genes associated with puQTL (puQTL genes) (310/1,927) did not overlap with those associated with eQTL (eQTL genes) (Fig 1E) and 42.2% best hit puQTL variants (1,095/2,592) and 40.9% LD-expanded puQTL variants ($r^2 > 0.8$) (1,061/2592) were not mapped as eQTL variants (S6 Fig), suggesting that puQTL mapping enables the identification of overlooked variant–gene associations. The difference in gene level matching and variant level matching indicates that a proportion of puQTL genes are associated with distinct loci from eQTL genes. The 16.1% unique puQTL genes included 23.5% single-promoter genes (73/310), indicating that the puQTL signals identified in this study might include false positive signals. As mentioned above, only junction reads spanning the first and second exons are used to estimate promoter activity, reducing the accuracy of the promoter activity estimation and the power of mapping QTL. Indeed, we found that 16.1% of puQTL genes not overlapping with eQTL genes have lower promoter activities than those overlapping with eQTL genes (S7 Fig). Thus, this result suggests that we have a limitation in mapping puQTL associated with lowly expressed promoters.

We also performed fine-mapping of puQTL to estimate candidate causal variants using the SuSiE software [34,35]. We identified 424 variants with a high causality (PIP > 0.9) and found that two of the credible causal variants were SVs (967_HG00733_ins and 28764_HG02059_ins), suggesting their independent roles in the perturbation of promoter usage.

Unlike eQTL, the puQTL analysis has the advantage of predicting associations between promoters and genomic variations. We expected that puQTL associated with multi-promoter genes would target the nearest promoter. However, the proportion of puQTL targeting the nearest promoters in all puQTL associated with multi-promoter genes was 29.1% (677/2,325) (Fig 1F). This result suggests that the effects of genetic factors on promoter usage are not easy

to dissect by simply focusing on genomic distances, which emphasizes the importance of puQTL mapping.

## puQTL are enriched in active promoters and enhancers

Next, we sought to examine the epigenetic contexts of puQTL to understand their functional mechanisms in gene regulation. We found that active histone marks for transcription (H3K4me3, H3K27ac, and H3K9ac) and ATAC-seq signals, which represent the open chromatin structure, were concentrated around puQTL, as well as eQTL (Fig 2A). Focusing on the 16.1% unique puQTL variants, they show relatively weaker enrichment of active epigenetic marks (H3K4me3, H3K27ac, and H3K9ac) than eQTL-overlapping puQTL variants (S8 Fig). This result indicates a possibility that unique puQTL variants tend to be located in lowly activated regulatory regions, or include false positive signals probably caused by limited power for the detection of lowly-expressed promoters. We assessed the statistical significance of the enrichment in epigenetic features based on permutation tests using the QTLtools fenrich module. The results of this analysis showed that puQTL and eQTL were significantly enriched in active histone marks, open chromatin regions detected by ATAC-seq, and TF footprints (Fig 2B and 2C). We confirmed that puQTL that were discovered without PEER factors as covariates were not enriched in transcriptionally active regions (S9 Fig), whereas those discovered using 25 PEER factors were enriched in those regions (Fig 2B). This result suggests that, although the greatest number of puQTL was detected when we used no PEER factors (S3B Fig), the puQTL discovered using 25 PEER factors were more credible than those identified without PEER factors. We also tested if the sets of QTL are enriched in regulatory elements using the chromatin states data of the GM12878 cell line defined by ChromHMM [43]. We found that both puQTL and eQTL were significantly enriched in upstream promoter regions (2_PromU) and active enhancer elements (13_EnhA1) at the 5% FDR level, implying the functional roles of the genomic variations in gene regulation via *cis*-regulatory elements, such as promoters and enhancers (Fig 2D and 2E). These results showed that the general features of chromatin states are similar between puQTL and eQTL.

## puQTL-associated gene classification

Next, we classified genes associated with puQTL into four groups in a similar way as that described by a previous study [12] (Fig 3A). Group-1 included 212 genes with a single promoter. We found that 65.6% puQTL genes having a single promoter (139/212) are also eQTL genes, but 34.4% (73/212) genes are not. This is probably due to the limited power to estimate promoter activity. Among the multi-promoter genes, 1,626 genes harboring only one promoter associated with puQTL were classified into group-2. Considering effect direction, which is a slope in the linear regression analysis (β), group-3 included 32 genes associated with puQTL with opposite effects on distinct promoters, and group-4 included 57 genes associated with puQTL with concordant effects on distinct promoters (Fig 3B). We also performed the same classification for 310 puQTL genes not overlapping eQTL genes. We found that 237 puQTL genes harbor multiple promoters and 225 puQTL genes have only one promoter associated with puQTL, while only 12 genes have multi-promoters associated with puQTL. This result indicates that a possible reason why 16.1% puQTL genes did not overlap with eQTL genes is that the genes have lowly-expressed puQTL-associated promoters that do not account for a large proportion of total gene expression. Based on this classification, we unveiled the detailed mechanisms of gene regulation by genomic variations.

We found an illustrative puQTL that was overlooked by annotation-based QTL analyses. Among the puQTL genes in group-4, the *TMEM200A* gene (Fig 3C) encodes a member of the

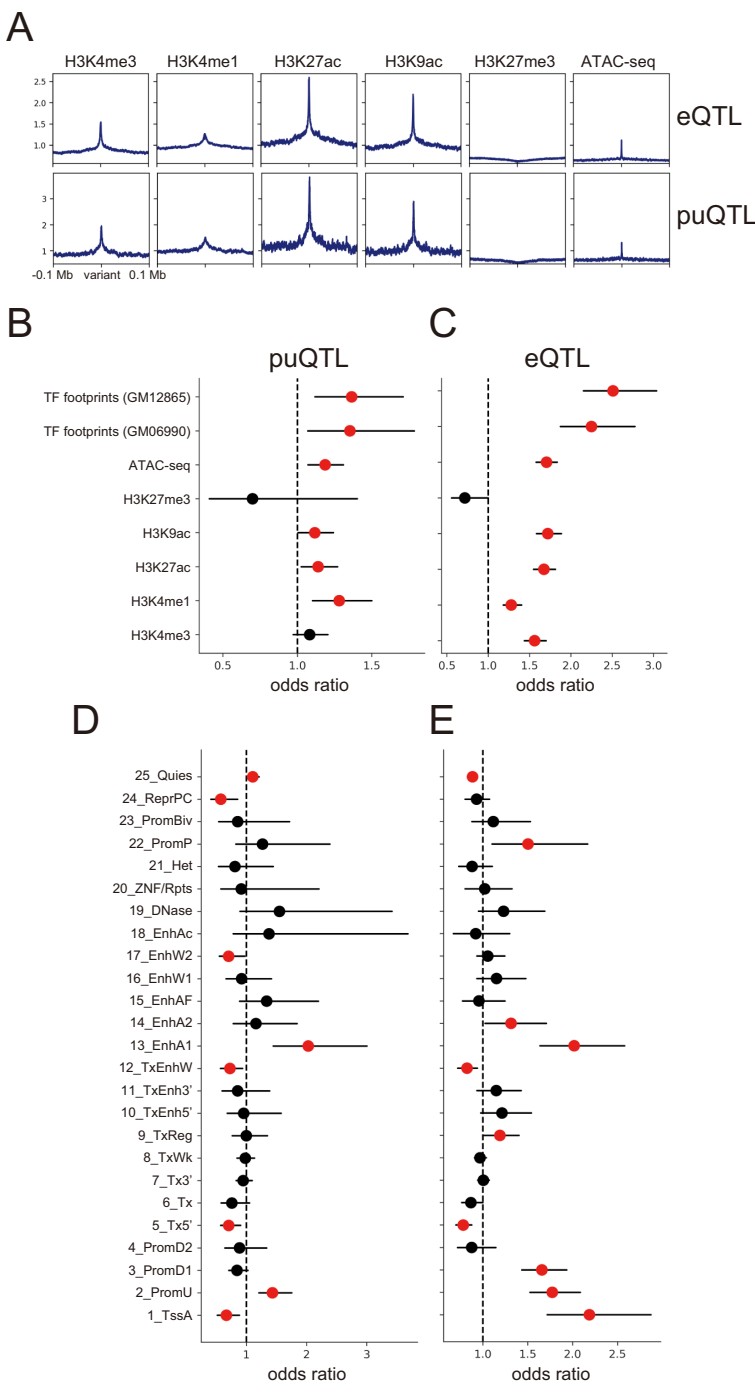

Figure 2 (Kubota and Suyama)

**Fig 2. Enrichment of epigenetic features.** (A) Aggregation plots of histone mark ChIP-seq and ATAC-seq signals for the 100 kb regions flanking puQTL best hit variants. (B–E) Enrichment of the peaks of histone mark ChIP-seq and transcription factor footprints at puQTL (B) and eQTL (C), and enrichment of the chromatin states defined by ChromHMM at puQTL (D) and eQTL (E). The red dots represent significant enrichment at the 5% FDR level, and the bars indicate 95% confidence intervals.

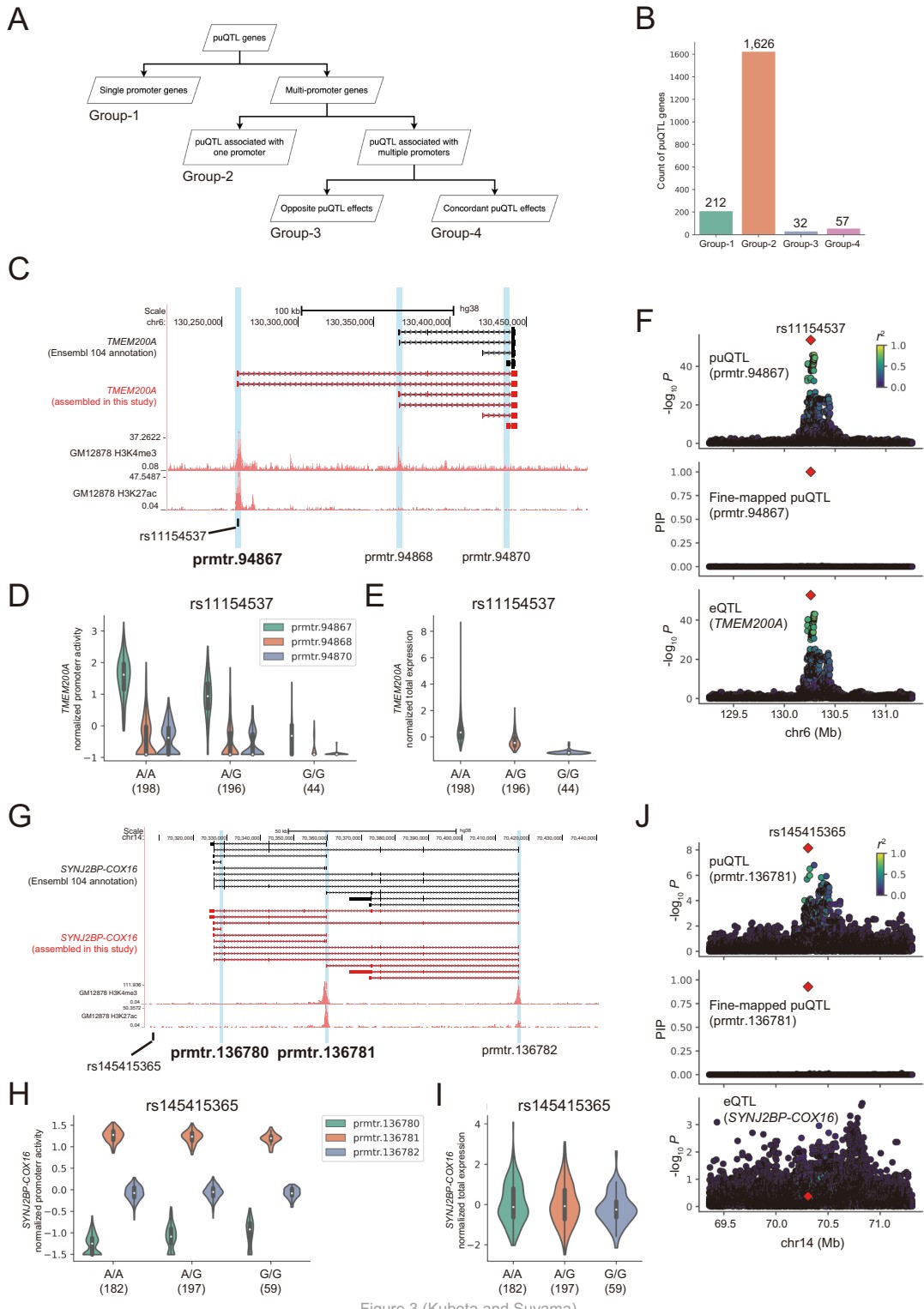

**Fig 3. puQTL-associated gene classification.** (A) The procedure used for puQTL gene classification. (B) Count of each group of puQTL genes. (C) The *TMEM200A* gene locus. The structures of the *TMEM200A* gene in the Ensembl 104 annotation and assembled in this study are presented in black and red, respectively, with ENCODE GM12878 H3K4me3 and H3K27ac ChIP-seq signals. The vertical blue bars indicate the location of active promoters. The black bar indicates the location of the rs11154537 variant. (D, E) Comparison of the promoter activities (D) and total expression levels (E) of the *TMEM200A* gene

among the rs11154537 genotypes. The numbers in parentheses indicate sample size. (F) The associations of puQTL, fine-mapped puQTL for prmtr.94867, and eQTL for *TMEM200A* are shown in the top, middle, and bottom panels. rs11154537 is plotted in a red diamond, and the colors indicate the *r*-squared values between rs11154537 and other variants. (G) The *SYNJ2BP-COX16* gene locus. (H, I) The promoter activities (H) and total expression levels (I) of the *SYNJ2BP-COX16* gene were compared among the rs145415365 genotypes. (J) Associations of puQTL, fine-mapped puQTL for prmtr.136781, and eQTL for *SYNJ2BP-COX16* are shown in the top, middle, and bottom panels.

transmembrane (TMEM) protein family. This gene has three active promoters (prmtr.94867, prmtr.94868, and prmtr.94870). Notably, isoforms transcribed from prmtr.94867 were not described in the Ensembl gene annotation file and were newly added by transcript assembly in our pipeline; therefore, the previous annotation-based analysis could not detect this promoter. The H3K4me3 and H3K27ac ChIP-seq signals also supported the contention that prmtr.94867 is transcriptionally active in the GM12878 cell line (Fig 3C). We found that rs11154537 (A/G), which is a variant located in the first exon of the isoform transcribed by prmtr.94867, was mapped as puQTL associated with the activity of prmtr.94867 ($P = 1.96 \times 10^{-54}$, β = −0.99) and prmtr.94870 ($P = 8.88 \times 10^{-8}$, β = −0.38), but not with prmtr.94868 ($P = 5.40 \times 10^{-4}$, β = −0.25) (Fig 3D). This variant was also mapped as eQTL associated with the total expression level of the *TMEM200A* gene ($P = 1.93 \times 10^{-53}$, β = −0.99) (Fig 3E). In addition, the result of the puQTL fine-mapping showed that the PIP of rs11154537, which represents the estimated causality in the perturbation of the promoter activity, was 1.00 for prmtr.94867 (Fig 3F), strongly suggesting that this variant is causal of the perturbation of prmtr.94867 usage. Taken together, these results indicate that an alternative allele of rs11154537 somehow leads to the downregulation of the "hidden" major promoter activity (prmtr.94867), resulting in a decrease in the total expression level of the *TMEM200A* mRNA. This is an example of the successful interpretation of how genomic variations alter total gene expression levels. Moreover, this example supports the conclusion that our computational pipeline is sufficiently powerful for detecting hidden active promoters associated with genomic variations, which were not included in eQTL and puQTL analyses based on standard annotation files.

Two puQTL genes associated with fine-mapped SVs were included in group-2. One of them is the *IVNS1ABP* gene, which encodes the influenza virus NS1A-binding protein. This gene was actively transcribed from one promoter, prmtr.86427 (S10A Fig). We found that 967_HG00733_ins, a genomic insertion of 34 bp, was located in the 90 kb upstream region of the gene and mapped as puQTL associated with the activity of prmtr.86427 ($P = 1.71 \times 10^{-12}$, β = 0.48) (S10B Fig). This SV was also mapped as eQTL associated with the total expression level of the *IVNS1ABP* gene ($P = 4.61 \times 10^{-56}$, β = 0.96) (S10C Fig). The chromatin interactions detected using H3K27ac HiChIP data of the GM12878 cell line [37] also support the functional connections between the SV and prmtr.86427 (S10A Fig). The PIP of 967_HG00733_ins was 0.99 for prmtr.86427 (S10D Fig). These results demonstrated that the intergenic 34 bp insertion upregulates the activity of one promoter via a 90 kb-range enhancer, thus increasing the total expression level of the *IVNS1ABP* mRNA. *RSPH1* was another puQTL gene associated with a fine-mapped SV. This gene encodes a radial-spoke-head protein and is transcribed from two distinct promoters (prmtr.70068 and prmtr.70069) (S10E Fig). Notably, the isoforms transcribed from prmtr.70068 were newly detected by transcript assembly in our pipeline, as in the case of the *TMEM200A* gene (Fig 3C). A 166 bp insertion located within the first exon of the isoform transcribed from prmtr.70068, 28764_HG02059_ins, was mapped as puQTL associated with the activity of prmtr.70068 ($P = 2.75 \times 10^{-45}$, β = −0.92), but not with prmtr.70069 ($P = 0.091$, β = −0.12) (S10F Fig). This SV was also mapped as eQTL associated with the total expression level of the *RSPH1* gene ($P = 6.77 \times 10^{-44}$, β = −0.91) (S10G Fig). Moreover, the PIP of 28764_HG02059_ins was 1.00 for prmtr.70068 (S10H Fig), strongly

suggesting that this exonic SV somehow leads to the downregulation of the promoter activity and a decrease in total *RSPH1* expression. These examples showed that our pipeline could identify cases in which SVs drive the dysregulation of promoter activities.

We also found several interesting examples of promoter usage changes without a gene expression change in group-3. One example was associated with the *SYNJ2BP-COX16* gene, which harbors three active promoters (prmtr.136780, prmtr.136781, and prmtr.136782) (Fig 3G). rs145415365 (A/G), a variant located in the downstream intergenic region of this gene, was mapped as puQTL associated with the activity of two promoters with opposite effects ($P = 1.67 \times 10^{-10}$, β = 0.44 for prmtr.136780; $P = 7.02 \times 10^{-9}$, β = −0.40 for prmtr.136781), but not with another promoter ($P = 0.52$, β = 0.045 for prmtr.136782) (Fig 3H). Notably, this variant was not associated with the total expression level of the *SYNJ2BP-COX16* gene ($P = 0.21$, β = −0.087) (Fig 3I), and the results of fine-mapping revealed a PIP of 0.93 for prmtr.136781 (Fig 3J), indicating the causality of this variant in the perturbation of promoter usage. We found another example of puQTL that was not mapped as eQTL in the *MUC12-AS1* gene. This gene harbors three active promoters (prmtr.99696, prmtr.99698, and prmtr.99699), two of which (prmtr.99696 and prmtr.99699) were not described in the Ensembl gene annotation file and were newly detected by our computational pipeline (S11A Fig). rs10229453 (T/C), a variant located in the intronic region of this gene, was mapped as puQTL associated with all promoters, but their effect directions were opposite ($P = 2.25 \times 10^{-56}$, β = −0.95 for prmtr.99696; $P = 5.02 \times 10^{-20}$, β = 0.60 for prmtr.99698; $P = 4.02 \times 10^{-9}$, β = 0.40 for prmtr.99699) (S11B Fig). This variant was not associated with the total expression level of the *MUC12-AS1* gene ($P = 0.21$, β = −0.087) (S11C Fig). These associations had not been detected by standard eQTL studies, which emphasizes the importance of our puQTL analysis for the discovery of new variant–gene associations.

## puQTL analyses enable novel interpretations of GWAS associations

To assess the involvement of puQTL in complex traits, we performed a comparative analysis of puQTL and GWAS lead variants. First, we applied the regulatory trait concordance (RTC) [52] method implemented in QTLtools to the set of puQTL and all GWAS lead variants curated by the GWAS Catalog. We found 1,690 puQTL, including 61 variants not mapped as eQTL, with a high-confidence concordance threshold (RTC > 0.9) for at least one trait. The set of puQTL included 166 credible causal variants (PIP > 0.9), implying the functional connections between promoter usage and traits. We found that a fine-mapped variant, rs2382817 (A/C) (PIP = 0.96), can be causal regarding the risk of inflammatory bowel diseases (IBDs), including ulcerative colitis and Crohn's disease. Previous studies have reported that the alternative allele of rs2382817 (rs2382817-C) is protective against IBD risk [53–55]. The variant was mapped as puQTL associated with two active promoters of the *TMBIM1* gene ($P = 3.50 \times 10^{-68}$, β = 1.03 for prmtr.64998; $P = 1.71 \times 10^{-9}$, β = 0.41 for prmtr.65000) (Fig 4A and 4B), and was also mapped as eQTL associated with the total gene expression ($P = 1.62 \times 10^{-41}$, β = 0.85) (Fig 4C). In addition, we performed the colocalization analysis implemented in the coloc software [47] to test if a causal variant can drive both puQTL and GWAS association. We found that a GWAS association for IBD [53] showed a high colocalization probability with the puQTL association for prmtr.64998 (PP4 = 0.90) and with the eQTL association for the *TMBIM1* gene (PP4 = 0.90) (Fig 4D). Taken together, these results demonstrate that an alternative allele of rs2382817 (rs2382817-C) might upregulate the minor promoter activity (prmtr.64998) and lead to a total increase in *TMBIM1* gene expression, resulting in a reduction of IBD risk. The TMBIM1 protein, which is also known as RECS1, is located in membranous compartments, including lysosomes, endosomes [56], and the Golgi

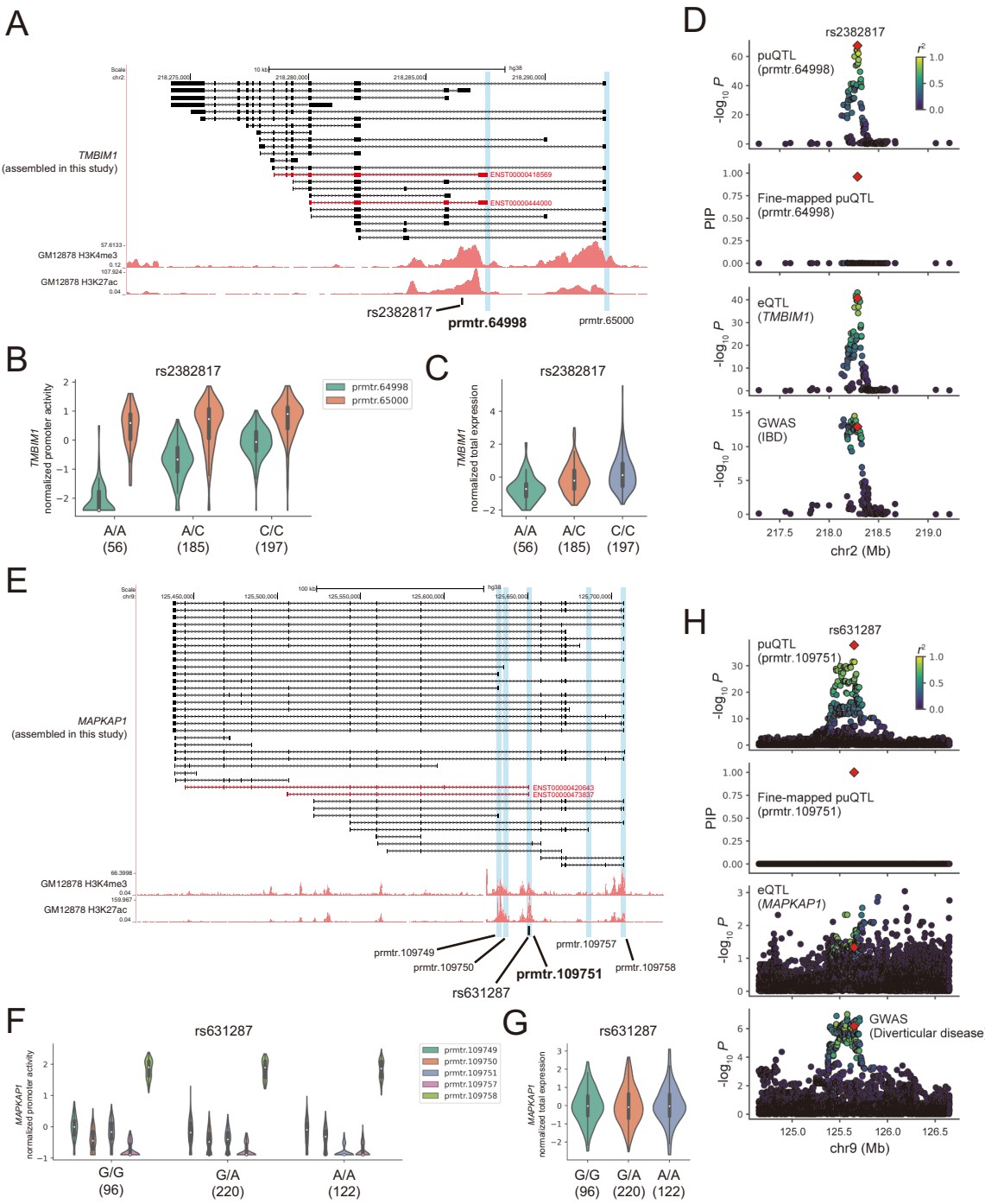

**Fig 4. puQTL and GWAS associations.** (A) The *TMBIM1* gene locus. Structures of the *TMBIM1* assembled in this study are presented in black with ENCODE GM12878 H3K4me3 and H3K27ac ChIP-seq signals. The isoforms transcribed from prmtr.64998 are shown in red. The vertical blue bars indicate the location of active promoters. The black bar indicates the location of the rs2382817 variant. (B, C) Comparison of the promoter activities (B) and total expression levels (C) of the *TMBIM1* gene among the rs2382817 genotypes. The numbers in parentheses indicate sample size. (D) The associations of puQTL, fine-mapped puQTL for prmtr.64998, eQTL for *TMBIM1*, and GWAS of inflammatory bowel disease (IBD) are shown from the top to bottom panels. rs2382817 is plotted in a red diamond, and the colors indicate *r*-squared values between rs2382817 and other variants. (E) The *MAPKAP1* gene locus. The structures of the *MAPKAP1* gene assembled in this study are presented in black, and the isoforms transcribed from prmtr.109751 are shown in red. (F, G) Comparison of the promoter activities (F) and total expression levels (G) of the *MAPKAP1* gene among the rs631287 genotypes. (H) The associations of puQTL, fine-mapped puQTL for prmtr.109751, eQTL for *MAPKAP1*, and GWAS of diverticular disease are shown from the top to bottom panels.

apparatus [57], and plays a protective role in Fas-mediated apoptosis by reducing Fas expression at the plasma membrane [58]. This is an illustrative example that we can clearly interpret the perturbed promoter in the context of disease risk.

Importantly, our pipeline was able to demonstrate that puQTL enables novel findings of variant–gene–trait associations. We found that the rs631287 (G/A) variant mapped as puQTL associated with one promoter (prmtr.109751) of the *MAPKAP1* gene ($P = 1.64 \times 10^{-38}$, β = −0.80), but not associated with other promoters ($P = 0.012$, β = −0.17 for prmtr.109748; $P = 0.54$, β = 0.042 for prmtr.109750; $P = 0.57$, β = −0.039 for prmtr.109757; and $P = 0.015$, β = 0.16 for prmtr.109758) (Fig 4E and 4F). The prmtr.109751 is one of the minor promoters of the *MAPKAP1* gene and does not account for a large portion of the total expression; therefore, rs631287 was not mapped as eQTL for the gene ($P = 0.046$, β = 0.14) (Fig 4G). We found that a GWAS association of diverticular disease [59], which is a condition that involves small pockets called diverticula that form from the wall of the colon, showed high colocalization probability with the puQTL association for prmtr.109751 (PP4 = 0.90); however, we observed no colocalization with the eQTL association for the *MAPKAP1* gene (PP4 = 0.091) (Fig 4H), indicating that standard eQTL studies cannot detect this variant–gene–trait association. The *MAPKAP1* gene, also known as *SIN1*, encodes the evolutionarily conserved protein Sin1, which is an essential component of the mammalian TOR complex 2 (mTORC2) [60,61]. The full-length *MAPKAP1* mRNA is translated into a protein that contains three domains; i.e., N-terminal domain (NTD), conserved region in middle domain, and Pleckstrin-homology domain (SIN1_HUMAN) (S12A and S12B Fig). However, we found that one isoform transcribed by prmtr.109751 (ENST00000420643) was predicted to be translated into an N-terminally truncated protein lacking the NTD sequence (BIAMA5_HUMAN) (S12A and S12C Fig). Previous studies have shown that the NTD of the Sin1 protein is important for its interaction with components of mTORC2, such as Rictor or MEKK2 [62,63]. The NTD-lacking isoform of Sin1, called Sin1δ, has been reported as not being responsible for the mTORC2 assembly and most of the mTORC2 functions [64]. Taken together, these results suggest that, although the precise function of NTD-lacking Sin1 is unclear, the variation of NTD-lacking Sin1 expression associated with the rs631287 genotype may underlie the risk of diverticular disease via mTORC2-independent mechanisms. Overall, these results emphasize that our puQTL analysis could provide new insights into associations between variants and complex traits, which were overlooked by previous eQTL analyses.

## Discussion

Here, we report a new computational method for discovering genomic loci associated with promoter usage that was constructed by leveraging conventional RNA-seq datasets without additional costs. Our analysis provided insights into the regulation of promoter activity by short and long genomic variations. The combined analysis of transcript assembly and promoter activity estimates enabled us to identify the perturbation of "hidden" promoters, which were overlooked by most annotation-based analyses. We performed eQTL mapping using the improved promoter annotations as well as puQTL mapping (S2 Fig), thus we can claim that the improved sensitivity is due to puQTL mapping itself. In addition, our integrated analysis of puQTL with GWAS led to novel interpretations of the molecular mechanisms of complex traits. This study emphasizes the importance of puQTL analysis in understanding gene regulation and common disease pathogenesis. Although we have a limitation in estimating activities of lowly expressed promoters, we believe that our pipeline is robust when mapping QTL associated with moderately or highly expressed promoters. Our method requires only genotype and RNA-seq data and, thus, can be applied to other extensive datasets provided by

international consortia, such as the GTEx project [2]. Identifying puQTL in various tissues using the GTEx datasets will help elucidate the tissue-specific regulation of promoter activities by genomic variations and promoter-usage-associated disease mechanisms in clinically relevant tissues. In this study, we mapped 2,592 puQTL associations and 1,927 puQTL genes using 438 samples. As for GTEx tissues with >100 sample size, assuming that the power of puQTL mapping increases with increasing sample size, we expect that 674 ~ 4,177 puQTL associations and 501 ~ 3,106 puQTL genes per tissue would be mapped by our pipeline. Assuming that new associations would be identified at the same ratio with this study (16.1%), we expect that 80 ~ 500 genes per tissue would be identified as new associations.

Changes in promoter usage can lead to changes in the relative abundance of transcripts harboring distinct 5′ untranslated regions (5′ UTR), which can affect mRNA translation rates [65]. In fact, the 5′ UTR includes various functional structures and elements, such as internal ribosome entry sites, binding sites for non-coding RNAs and RNA-binding proteins, RNA modification sites, hairpins, pseudo-knots, and RNA G-quadruplexes. These structures and elements regulate the recruitment of eukaryotic initiation factors to the 5′ end of mRNAs, and affect the mRNA translation rates, indicating the possibility that perturbation of promoter activity associated with genomic variations may result in changes in protein abundance. However, our analysis had a limitation: we focused exclusively on mRNA expression levels, rather than protein expression; therefore, we were unable to assess how promoter changes affect translation regulation. Previous studies have reported the association between genomic loci and protein expression levels (protein QTL: pQTL) [66–68]; thus, by integrating the puQTL and pQTL data obtained from matched samples, we were able to discover promoter-change-associated protein expression changes. It is well known that protein expression levels are often uncorrelated with corresponding mRNA expression levels [69–72], suggesting that it may become possible to interpret why protein levels change in the absence of a total mRNA expression change by focusing on promoter perturbation.

Promoter usage and splicing are closely interrelated. A recent study reported that splicing of internal exons promotes the activation of cryptic promoters, which is called exon-mediated activation of transcription starts (EMATS) [73]. Splicing factors interact with general TFs and recruit them to nearby weak promoters, resulting in promoter activation and an increase in transcriptional output [73]; however, the genetic factors that drive EMATS remain unexplored. In addition to gene expression, genomic variations also affect alternative splicing (splicing QTL: sQTL), thus implying that genomic variations affect the inclusion level of internal exons and lead to differential usage of nearby promoters in some genes. Although our analysis focused on the enrichment of genomic variations associated with promoter usage in promoters, enhancers, and TF binding sites (Fig 2), the combined analysis of puQTL and sQTL has the potential to identify genetic factors underlying splicing-mediated promoter activation. Our approach, which was developed to map puQTL using RNA-seq data, is valuable for exploring novel gene regulatory programs.

In this study, reference-guided transcript assembly using StringTie2 [28] and short-read RNA-seq reads of 438 samples enabled us to find "hidden" active promoters (Figs 3C, S7E, and S8A), highlighting the fact that the standard gene annotations provided by Ensembl are incomplete. It is because promoters expressed in a small proportion of the human population, which may be regulated by genomic variations, are likely to be overlooked when annotation files are constructed based on limited RNA-seq data. However, some major promoters, like the *TMEM200A* example, somehow have been missing. We are not exactly sure why some highly expressed promoters are not described in the Ensembl annotation file (probably this reflects limitations of the Ensembl pipeline and manual curation by humans), but in general, we believe that we can improve sensitivity of isoform prediction by incorporating mapped reads

from as many samples as possible. The precise construction of gene annotations is essential to explore the regulatory mechanism of promoter activity. However, the construction of transcript structures using only short-reads has been challenging. Notably, the StringTie2 software also works with long-read RNA sequencing reads and shows high performance in transcript assembly [28]. A recent study generated a long-read RNA-seq dataset using the Oxford Nanopore Technologies platform from 88 samples from the GTEx project [74]. The leveraging of the GTEx short- and long-read RNA-seq dataset would improve our pipeline for mapping additional puQTL.

Taken together, the findings of this study suggest that our pipeline may be the current best practice for accurate RNA-seq-based puQTL mapping. Moving forward, by expanding this pipeline to extensive datasets in public databases, our strategy can help us further elucidate the genetic factors underlying transcriptional regulation and promote the novel discovery of the molecular mechanisms responsible for complex trait associations.

## Supporting information

**S1 Fig. Correlation of estimated promoter activities among three different methods.** Activity scores of each promoter of the GM12878 cell line are plotted with regression lines.
(PDF)

**S2 Fig. Pipeline for eQTL mapping.** 1KGP, the 1000 Genomes Project; MAF, Minor Allele Frequency; GTF, Gene Transfer Format.
(PDF)

**S3 Fig. Covariates for QTL analysis.** (A) Projection on the first two principal components of the normalized genotype data, labelled for populations. GBR, British in England and Scotland; FIN, Finnish in Finland; CEU, Utah residents (CEPH) with Northern and Western European ancestry; YRI, Yoruba in Ibadan, Nigeria; TSI, Toscani in Italia. (B, C) The number of puQTL (B) and eQTL (C) (nominal $P < 1.0 \times 10^{-5}$) identified (y-axis) versus the number of PEER factors used (x-axis). Red circles represent the number of PEER factors used for the following analysis.
(PDF)

**S4 Fig. eQTL analysis results.** (A) Quantile–quantile plot of $P$-values. The nominal pass results of chromosome 22 are plotted and a red line indicates expected $P$-values under the null hypothesis. (B) Distribution of the distance of eQTL best hit variants from the target promoters.
(PDF)

**S5 Fig. Examples of puQTL.** (A) The *TTC23* gene locus. Structures of the *TTC23* assembled in this study are shown with ENCODE GM12878 H3K4me3 and H3K27ac ChIP-seq signals. A vertical blue bar indicate the location of an active promoter, prmtr.35339. A black bar indicates the location of a variant rs8028374. (B) Comparison of the promoter activities of the *TTC23* gene among rs8028374 genotypes. The numbers in parentheses indicate sample size. (C) The *DENND2D* gene locus. (D) Comparison of the promoter activities of the *DENND2D* gene among rs35430374 genotypes.
(PDF)

**S6 Fig. Overlaps of puQTL and eQTL associations at the variant level.** (A) Propotion of puQTL associations mapped as eQTL. (B) Propotion of puQTL associations mapped as eQTL when expanding linkage disequilibrium ($r^2 > 0.8$).
(PDF)

**S7 Fig. Promoter activity of puQTL genes.** The y-axis represents median values of promoter activity of puQTL genes mapped as eQTL genes (green) and not mapped as eQTL genes (orange) in 438 individuals.
(PDF)

**S8 Fig. Enrichment of epigenetic features of puQTL variants.** Aggregation plots of histone mark ChIP-seq and ATAC-seq signals for 100 kb regions flanking puQTL best hit variants. Green and blue lines represent aggregated signals for puQTL variants mapped as eQTL and those not mapped as eQTL, respectively.
(PDF)

**S9 Fig. Enrichment of puQTL identified without PEER factors in epigenetic features.** Enrichment of peaks of histone mark ChIP-seq and transcription factor footprints. Red dots represent significant enrichment at the 5% FDR level and bars show 95% confidence intervals.
(PDF)

**S10 Fig. Structural variants associated with promoter usage.** (A) The *IVNS1ABP* gene locus. Structures of the *IVNS1ABP* assembled in this study are represented in black with ENCODE GM12878 H3K4me3, H3K4me1, and H3K27ac ChIP-seq signals and H3K27ac HiChIP chromatin interactions. Vertical blue bar indicates the location of an active promoter prmtr.86427. A black bar indicates the location of a structural variant 967_HG00773_ins. (B, C) Comparison of the promoter activities (B) and total expression levels (C) of the *IVNS1ABP* gene among 967_HG00773_ins genotypes. The numbers in parentheses indicate sample size. (D) Associations of puQTL, fine-mapped puQTL for prmtr.86427, and eQTL for *IVNS1ABP* are shown in the top, middle, and bottom panel. 967_HG00773_ins is plotted in a red diamond and colors indicate r-squared values between 967_HG00773_ins and other variants. (E) The *RSPH1* gene locus. Structures of the *RSPH1* in the Ensembl 104 annotation and assembled in this study are represented in black and red, respectively with ENCODE GM12878 H3K4me3 and H3K27ac ChIP-seq signals. Vertical blue bars indicate the location of an active promoters. A black bar indicates the location of a structural variant 28764_HG02059_ins. (F, G) Comparison of the promoter activities (F) and total expression levels (G) of the *RSPH1* gene among 28764_HG02059_ins genotypes. (H) Associations of puQTL, fine-mapped puQTL for prmtr.70068, and eQTL for *RSPH1* are shown in the top, middle, and bottom panel. 28764_HG02059_ins is plotted in a red diamond and colors indicate r-squared values between 28764_HG02059_ins and other variants.
(PDF)

**S11 Fig. puQTL associated with distinct promoters of the *MUC12-AS1* gene with opposite effects.** (A) The *MUC12-AS1* gene locus. Structures of the *MUC12-AS1* in the Ensembl 104 annotation and assembled in this study are represented in black and red, respectively, with ENCODE GM12878 H3K4me3 and H3K27ac ChIP-seq signals. Vertical blue bars indicate the location of active promoters. A black bar indicates the location of a variant rs10229453. (B, C) Comparison of the promoter activities (B) and total expression levels (C) of the *MUC12-AS1* gene among rs10229453 genotypes. The numbers in parentheses indicate sample size. (D) Associations of puQTL, fine-mapped puQTL for prmtr.99696, and eQTL for *MUC12-AS1* are shown in the top, middle, and bottom panel. rs10229453 is plotted in a red diamond and colors indicate r-squared values between rs10229453 and other variants.
(PDF)

**S12 Fig. Comparison of the Sin1 proteins translated from distinct transcripts.** (A) Multiple sequence alignment of two isoforms of the Sin1 (MAPKAP1) protein. Protein sequences of full-length Sin1 (SIN1_HUMAN) and NTD-lacking Sin1 (BIAMA5_HUMAN) are aligned in top and bottom rows, respectively. Asterisks (*) indicate positions of matched residues. Red, black, and blue rectangles indicate sequences of N-terminal domain, conserved region in middle domain, and Preckstrin-homology domain, respectively. (B, C) Protein structures of full-length Sin1 (SIN1_HUMAN) (B) and NTD-lacking Sin1 (BIAMA5_HUMAN) (C) predicted by AlphaFold2 with colors representing per-residue confidence score. Dotted circles indicate N-terminal domain.
(PDF)

**S1 Table. Samples from the GEUVADIS Project analyzed in this study.**
(XLSX)

## Acknowledgments

We thank all members of Mikita Suyama's laboratory for their valuable discussions. The authors would like to thank Enago (www.enago.jp) for the English language review.

## Author Contributions

**Conceptualization:** Naoto Kubota.

**Data curation:** Naoto Kubota.

**Formal analysis:** Naoto Kubota.

**Investigation:** Naoto Kubota.

**Methodology:** Naoto Kubota.

**Project administration:** Mikita Suyama.

**Software:** Naoto Kubota.

**Supervision:** Mikita Suyama.

**Validation:** Naoto Kubota, Mikita Suyama.

**Visualization:** Naoto Kubota.

**Writing – original draft:** Naoto Kubota.

**Writing – review & editing:** Naoto Kubota, Mikita Suyama.

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
