## [Decision Letter · Decision Letter 0]

17 May 2022

Dear Dr. Suyama,

Thank you very much for submitting your manuscript "Mapping of promoter usage QTL using RNA-seq data reveals their contributions to complex traits" for consideration at PLOS Computational Biology.

As with all papers reviewed by the journal, your manuscript was reviewed by members of the editorial board and by several independent reviewers. In light of the reviews (below this email), we would like to invite the resubmission of a significantly-revised version that takes into account the reviewers' comments.

Two reviewers have thoroughly considered the merits of this submission. Both are enthusiastic about the topic area and the potential of the approach as described. However, both have major concerns with whether the current presentation is comprehensive, fully methodologically informative, and accurately framed in terms of the advancement achieved. Therefore, the manuscript cannot be accepted for publication at this time. A revision that fully addresses all points raised by reviewers could be considered. Substantial revision would be required to ensure that, in particular, reviewer 2's points 1 and 2 are fully addressed, alongside all other points by reviewer 2 and all other points by reviewer 1 (some of which are overlapping).

We cannot make any decision about publication until we have seen the revised manuscript and your response to the reviewers' comments. Your revised manuscript is also likely to be sent to reviewers for further evaluation.

Sincerely,

Jeffrey Peter Townsend, Ph.D.

Guest Editor

PLOS Computational Biology

Ilya Ioshikhes

Deputy Editor

PLOS Computational Biology

Two reviewers have thoroughly considered the merits of this submission. Both are enthusiastic about the topic area and the potential of the approach as described. However, both have major concerns with whether the current presentation is comprehensive, fully methodologically informative, and accurately framed in terms of the advancement achieved. Therefore, the manuscript cannot be accepted for publication at this time. A revision that fully addresses all points raised by reviewers could be considered. Substantial revision would be required to ensure that, in particular, reviewer 2's points 1 and 2 are fully addressed, alongside all other points by reviewer 2 and all other points by reviewer 1 (some of which are overlapping).

Reviewer's Responses to Questions

**Comments to the Authors:**

Reviewer #1: Kubota and Suyama present a manuscript describing a new method for the identification of associations between the presence of genetic variants and the activity of promoters using RNA-Seq data. This method allows more precision during the functional interpretation of GWAS variants, as well as to identify some signals that traditional eQTL cannot detect with its bulk-based approach to quantify gene expression. The authors assess the value of this method, puQTL, using RNA-Seq and variation from cancer cell lines, comparing the results with CAGE-based methods, and describe supporting facts and cases at functional level. In my opinion, this is an interesting and promising method to start advancing in the functional interpretation of GWAS variants, and to start overcoming common simplifications and limitations of current eQTL analyses. The manuscript is well written and clear.

I have though some concerns, mostly regarding the evaluation of the method and its components:

- In page 9, the authors claim that proActive estimations of promoter activity were “correlated more strongly” with CAGE, than the kallisto method. But they do not describe the level and type of correlation. Even assuming the limitations of the CAGE method, it is necessary to better describe the scope of proActive, as it is essential to understand which promoters are, or are not considered inside puQTL.

- I think it is important to understand better the nature of the 16.1% of puQTL, not captured by eQTL, as it constitutes the main benefit of this method. Around this comparison, there are several questions that need to be answered and also discussed. For example:

- is the puQTL vs eQTL signals comparison based on variant, or on LD-block?.

- How many associations match at variant level and at LD level?

- Is the absence of promoter definition by proActive, the reason for the different number of overall signals (2592 vs 18205)?

- What is the number of input genes/expression considered for puQTL and eQTL?

- Does the 16.1% correlate more or less with epigenetics signals, than the other 83.9% puQTL overlapping eQTLs? As for chromatin non-associated puQTLs, is this 16,1% of new signals also more enriched in non-PEER?

- As one would expect, is the 16% of puQTL unique signals covering mostly multi-promoter genes? If there are single-promoter genes within this 16%, any idea of why they were not detected by eQTL?

- As for the TMEM200A example, what is the overall limitation expected from the Ensemble file, and how this could be improved. Could we improve sensitivity by being more inclusive when predicting isoforms? Please discuss.

- In my opinion, the authors should emphasize in the discussion (and backup with numbers in results) the added value of this methodology, beyond single representative cases. For example, how many new associations should we expect in the case we would apply this method to GTEX? What are the kinds of genes where we expect a gain? Are the associations with multipromoter genes expected to co-localize more with quantitative traits, vs binary traits?

Minor:

Page 9, “Ensemble gene annotation file” is vague. Which type of genes are considered? What is in the GTF file? Coding and not coding? This is not clear from the main text.

Pge 10, please further clarify the example described for rs802…. and rs354… . What do the authors try to demonstrate? The absence of intragenic signals? Now, it is difficult to understand.

Reviewer #2: This study by Kubota and Suyama seeks to map QTLs associated with variation in promoter usage in the GEUVADIS population dataset. This is a very important undertaking to understand the mechanisms of gene expression and map genetic variants that directly regulate variation in transcription initiation. In general, the study is fairly comprehensive and well written. However, I have a few major concerns to be addressed:

(1) It is unclear to me that this manuscript presents a unified computational framework with which to identify puQTL. While the study uses a different software to quantify promoter usage, this software itself has been published elsewhere, and the rest of the pipeline involves a standard set of bioinformatic QC, filtering, and data cleaning up steps that are not novel in and of themselves. The authors should make it more clear why they refer to this as a new framework (to me, indicating an algorithm, set of algorithms, or innovative way of thinking of or implementing a computational problem or pipeline)

(2) I am still a bit unclear which molecular metric or phenotype is being mapping. Specifically, are the authors mapping isoforms levels (by TPMs), differential promoter usage (ratios of promoter usage within a gene), or some other metric? The choice of metric drastically affects the interpretation of the QTLs and the expected overlap with eQTLs, etc. For instance, the authors state that 212 puQTL genes have a single promoter – are all of these puQTL then also eQTL (since presumably this is mapping isoform levels)?

(3) The authors state that 16.1% of genes associated with puQTLs did not overlap with those associated with eQTLs. It would be nice to perform further analyses on characteristics of these genes to identify either technical or biological features that differentiate these genes. Do they tend to have multiple promoters whose usage balances out to maintain constant gene expression levels? Is there an bias towards lower or higher expressed genes in this group and thus either QTL methods is underpowered to identify QTL?

(4) Most of the examples of novel QTLs that are identified only through puQTL methods rather than eQTL methods seem to be the result of the improved annotations conducted by the authors. This seems like a different biological message than that highlighted by the authors, which claims that mapping puQTLs itself uncovers novel regulatory genetic loci. However, instead, it seems like their pipeline does a better job annotating complete gene regions, leading to the identification of more QTL. This can be quantified by comparing how many more eQTL are identified using the improved promoter annotations. In general, it would be useful for the authors to show or speculate a bit more why they believe the puQTL associations were not detected by standard eQTL studies.

Minor:

(1) I know the method itself has been published elsewhere, but the authors should add a few sentences on exactly how proActiv identifies promoter activities and why these computational advances/advantages are best suited for their pipeline.

**Have the authors made all data and (if applicable) computational code underlying the findings in their manuscript fully available?**

Reviewer #1: Yes

Reviewer #2: Yes

PLOS authors have the option to publish the peer review history of their article (what does this mean?). If published, this will include your full peer review and any attached files.

Reviewer #1: No

Reviewer #2: No
---

## [Decision Letter · Decision Letter 1]

25 Jul 2022

Dear Dr. Suyama,

We are pleased to inform you that your manuscript 'Mapping of promoter usage QTL using RNA-seq data reveals their contributions to complex traits' has been provisionally accepted for publication in PLOS Computational Biology.

Best regards,

Jeffrey Peter Townsend, Ph.D.

Guest Editor

PLOS Computational Biology

Ilya Ioshikhes

Deputy Editor

PLOS Computational Biology

Reviewer's Responses to Questions

**Comments to the Authors:**

Reviewer #1: In my opinion, the authors have properly addressed and clarified all my original concerns (both major and minor) of the first version of the manuscript. The authors have also modified the new version of the ms accordingly, with which I agree.

**Have the authors made all data and (if applicable) computational code underlying the findings in their manuscript fully available?**

Reviewer #1: Yes

PLOS authors have the option to publish the peer review history of their article (what does this mean?). If published, this will include your full peer review and any attached files.

Reviewer #1: No

---

## [Editor Report · Acceptance letter]

23 Aug 2022

PCOMPBIOL-D-22-00463R1 

Mapping of promoter usage QTL using RNA-seq data reveals their contributions to complex traits

Dear Dr Suyama,

I am pleased to inform you that your manuscript has been formally accepted for publication in PLOS Computational Biology. Your manuscript is now with our production department and you will be notified of the publication date in due course.

With kind regards,

Zsofia Freund
